# Mid-Holocene pulse of thinning in the Weddell Sea sector of the West Antarctic ice sheet

Andrew S. Hein[1], Shasta M. Marrero[1], John Woodward[2], Stuart A. Dunning[2,3], Kate Winter[2], Matthew J. Westoby[2], Stewart P.H.T. Freeman[4], Richard P. Shanks[4] & David E. Sugden[1]

Establishing the trajectory of thinning of the West Antarctic ice sheet (WAIS) since the last glacial maximum (LGM) is important for addressing questions concerning ice sheet (in)stability and changes in global sea level. Here we present detailed geomorphological and cosmogenic nuclide data from the southern Ellsworth Mountains in the heart of the Weddell Sea embayment that suggest the ice sheet, nourished by increased snowfall until the early Holocene, was close to its LGM thickness at 10 ka. A pulse of rapid thinning caused the ice elevation to fall ∼400 m to the present level at 6.5–3.5 ka, and could have contributed 1.4–2 m to global sea-level rise. These results imply that the Weddell Sea sector of the WAIS contributed little to late-glacial pulses in sea-level rise but was involved in mid-Holocene rises. The stepped decline is argued to reflect marine downdraw triggered by grounding line retreat into Hercules Inlet.

[1] School of GeoSciences, University of Edinburgh, Drummond Street, Edinburgh EH8 9XP, UK. [2] Department of Geography, Northumbria University, Ellison Place, Newcastle upon Tyne NE1 8ST, UK. [3] School of Geography, Politics and Sociology, Newcastle University, Newcastle upon Tyne NE1 7RU, UK. [4] Scottish Universities Environmental Research Centre, Rankine Avenue, East Kilbride G75 0QF, UK. Correspondence and requests for materials should be addressed to A.H. (email: Andy.Hein@ed.ac.uk).

The West Antarctic ice sheet (WAIS) has undergone a complicated process of thinning and retreat since the last glacial maximum (LGM) with different parts of the ice sheet behaving in contrasting ways[1–5]. These complex ice dynamics affect our understanding of whether the WAIS was involved in late-glacial sea-level rise[6,7] and its contribution to sea-level rise in the Holocene[8–10]. One view of the overall trajectory of the WAIS since the LGM assumes a progressive loss of ice after about 15 ka. In some offshore locations bordering the western Antarctic Peninsula and the Amundsen Sea, most retreat had occurred by 9–10 ka (refs 11,12), while some coastal uplands emerged later at around 8 ka (ref. 13). In the Ross Sea, retreat of grounded ice was underway by ∼13 ka and continued until 2–3 ka (ref. 14). Ice volumes and rates of ice-mass loss in the Weddell Sea sector are uncertain. Indeed, there is a debate about whether the LGM ice was thin or thick and whether it extended out onto the Weddell Sea offshore shelf; the debate is reflected in contrasting interpretations based both on field observations[7,15–18] and ice-sheet models[1,3,19].

An alternative view based on the latest high-resolution dating of the WAIS Divide ice core is that the central ice dome was lower than present during the LGM and thickened in response to increased snowfall caused by late-glacial climatic warming[20,21]. Thickening subsequent to the LGM has also been supported by cosmogenic nuclide data from nunataks in the interior of West Antarctica and the Transantarctic Mountains that indicates the ice surface elevation was at its maximum at 12–7 ka (refs 22–24), and by a detailed set of radiocarbon ages demonstrating an advance of WAIS ice in the Ross Sea at 18.7–12.8 ka (ref. 25). Eventually, the interior thickening due to increased snowfall is overcome by rising global sea levels and ocean warming that melts coastal ice shelves and triggers thinning of outlet glaciers, initiating a wave of thinning that propagates to the interior of the ice sheet. Ice retreat and thinning along the Pacific margins of the Amundsen Sea coast and southern Antarctic Peninsula continues and is accelerating[26]. In contrast, glacial isostatic adjustment models suggest that in the Weddell Sea sector the ice could have thickened in the last 4–2 ka (ref. 27).

The Patriot, Independence and Marble Hills are situated in the southern Heritage Range, Ellsworth Mountains in the heart of the inner Weddell Sea embayment (Figs 1–3, Supplementary Fig. 1). The mountains are within 50 km of the grounding line, which marks the interface between the grounded ice sheet and the floating Filchner–Ronne Ice Shelf in Hercules Inlet; such a location is sensitive to changes in ice thickness in the wider Weddell Sea area[1,16,19]. At present, ice from the main WAIS dome summits, some 200–300 km to the northwest and west, respectively, flows around and between the massifs towards the grounding line. Blue-ice moraines form on the windswept ice surface at the foot of the mountains and remnants of earlier blue-ice moraines occur up to 650 m above the present ice surface[28]. Little-weathered blue-ice deposits extend up to 475 m above the ice surface and mark the decline in ice surface elevation since the LGM[7,29].

Here we present new evidence from the southern Heritage Range that allows us to constrain both the deglacial history of the ice sheet and its relationship to changing patterns of ice flow. Our evidence derives from detailed multi-isotope cosmogenic nuclide analyses tied closely to the geomorphology. We find geomorphological evidence indicating an easterly ice flow direction at the LGM that contrasts with the more northerly ice flow direction at present. Cosmogenic $^{10}$Be, $^{26}$Al and $^{36}$Cl exposure ages indicate the ice sheet was close (∼20 m below) to its LGM maximum position in the early Holocene at 10 ka; this suggests deglacial thinning began ∼5 ka later than previously thought[7]. After a slight decline in ice surface elevation, a pulse of rapid thinning caused the ice elevation to fall by ∼400 m at 6.5–3.5 ka. At the Patriot Hills, the ice elevation today is the same as at 3.5 ka. The results imply that the Weddell Sea sector of the WAIS contributed little to late-glacial pulses in sea-level rise but was involved in mid-Holocene rises. We link the late deglaciation of this part of the ice sheet to increased accumulation that offset marine downdraw until the early Holocene. Our geomorphological investigation indicates a change in ice-flow direction accompanied the pulse of thinning. Therefore, we argue that the mid-Holocene pulse of thinning was caused by marine downdraw triggered by grounding line retreat into Hercules Inlet.

## Results

**Geomorphology.** The deposits comprise a veneer of glacial debris, including glacially abraded and often striated perched

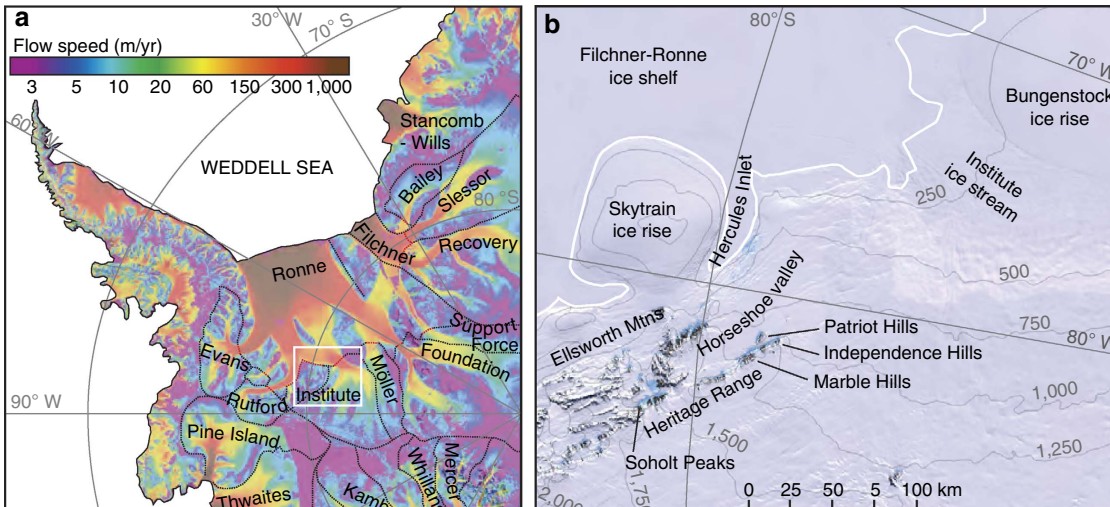

**Figure 1 | Location map showing the position of the Heritage Range in the heart of the Weddell Sea embayment.** (**a**) The Heritage Range in relation to ice-stream basins. The image shows satellite-derived surface ice-flow velocities of the Antarctic Ice Sheet from MEaSUREs[53], annotated to show the dominant ice streams and their catchment areas. The white box indicates the location of **b**. (**b**) The Heritage Range field site in relation to Hercules Inlet and Institute ice stream. The figure shows a MODIS and LIMA mosaic of Antarctica with prominent geographical features labelled. White line indicates the ASAID grounding line[54], thin grey lines are surface elevation contours at 250 m intervals from Bedmap2 (ref. 55).

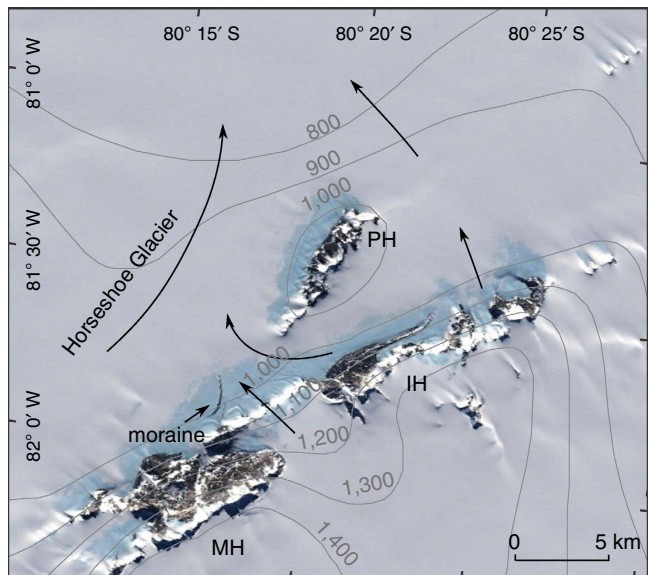

**Figure 2 | The southern Heritage Range field site.** The Google Earth imagery (Map data: U.S. Geological Survey) shows the location of the Patriot Hills (PH), Independence Hills (IH) and Marble Hills (MH). It also shows the location of blue-ice areas, drawn-out medial moraines between the MH and IH massifs, and current ice flow directions (black arrows). Surface contours are from Bedmap2 (ref. 55).

blocks, sometimes overlying buried glacier ice (Fig. 3a, Supplementary Fig. 2). This upper limit is at 475, 400 and 230 m above the present ice surface in the Marble, Independence and Patriot Hills, respectively. The altitudinal differences reflect the local topography with higher limits where the ice is constricted and lower limits where it spreads out in the lee of massifs. The gradient of the upper margin falls to the east and this is consistent with geomorphic evidence of eastward flow on the plateau surfaces of the Marble Hills and on the spurs of the northern flank of the Patriot Hills[28]. Eastward flow is confirmed by the carry of basic igneous erratics in the blue-ice moraines that originate in the Soholt Peaks area (Fig. 1b) 70 km to the west[30]. Eastward flow at the LGM is also consistent with the overall orientation of the blue-ice moraine extending some 10 km to the east of the Independence Hills (Fig. 3c). Here the addition of ice overspilling from the main WAIS dome displaces the blue-ice moraine and, as is consistent with eastward flow, forms a medial moraine that is progressively displaced from the mountain foot. Cosmogenic $^{10}$Be and $^{36}$Cl ages of 17–43 ka from large boulders on the blue-ice moraine ridge itself are consistent with the moraine reflecting ice flow at the LGM (Supplementary Fig. 1).

Present ice flow directions to Hercules Inlet in the north contrast with the eastward flow implied by the elevated blue-ice deposits. Outlet glaciers between Marble and Independence Hills flow northward into Horseshoe Glacier, as evidenced by the surface gradient, flow structures and the orientation of medial moraines between the massifs (Figs 2 and 3c). The western part of the Independence Hills blue-ice moraine is part of this flow and, unusually for blue-ice moraines, which form under compressive lateral flow toward the massif, it is broken up by a series of

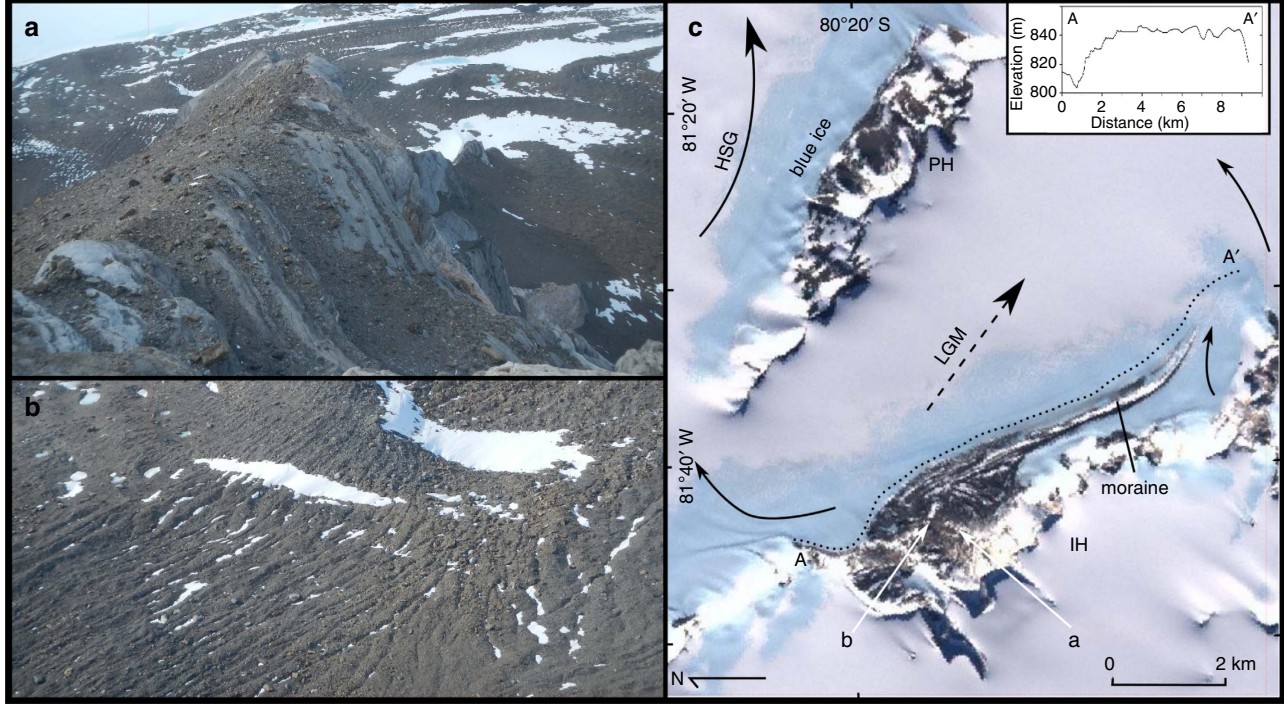

**Figure 3 | The Patriot and Independence Hills field site and evidence for ice-flow reversal.** (**a**) Perched blocks characteristic of the little-weathered deposits of the last glacial cycle. (**b**) Extensional crevasses within the blue-ice moraine at the western end of the Independence Hills are consistent with present northward flow implied by the northward dipping surface gradient at this location, and with flow structures in the ice. (**c**) MODIS and LIMA mosaic showing the Patriot Hills (PH) and the medial blue-ice moraine extending east from the Independence Hills (IH). Ice overspilling into the valley has progressively displaced the moraine away from the mountain foot as is consistent with former eastward flow. Current ice flow directions (black solid arrows) show flow has been diverted northward into Horseshoe Glacier (HSG). The inset shows the ice margin profile between A and A' with a northward dipping gradient to the west. The southerly dipping gradient near A' is into a zone of local ablation.

extensional crevasses oriented perpendicular to the massif; moreover, the overall surface elevation of the disrupted part of the moraine falls to the northwest (Fig. 3). These geomorphological characteristics show that in this location an eastward flow of ice has been reversed during the process of ice sheet thinning.

**Cosmogenic nuclide data.** We measured cosmogenic $^{10}Be$, $^{26}Al$ and $^{36}Cl$ in unweathered, glacially abraded cobbles resting on bedrock spurs in each massif and on the current ice margin (Figs 4 and 5; Supplementary Figs 1–4; Supplementary Data 1–3). Our data set comprises 71 cosmogenic nuclide dates, and we have included eleven samples previously collected from the same spurs[7]. Rock surface erosion is demonstrably negligible on most of the sampled clasts, and thus no erosion correction is applied. We assume the youngest sample at each elevation best represents the deposition age[2]. This is appropriate given our assessment of inheritance and predepositional exposure within a blue-ice moraine system. A full discussion on these topics and our data is included in the Methods section.

Exposure ages from at or near the upper limit of little-weathered erratics range from 49 to 10 ka in the Marble Hills, 26–10 ka in the Independence Hills and 39–9 ka in the Patriot Hills. A pulse of rapid thinning at 6.5–3.5 ka is evident from the similarity of exposure ages over a large elevation range at each massif. Abrupt thinning from near the upper limit in the Marble Hills began at $6.5 \pm 0.3$ ka. In the Independence Hills, the sudden drop in ice elevation approached within 35 m of the present ice level by $3.7 \pm 0.2$ ka, while in the Patriot Hills the present ice level was achieved by $3.5 \pm 0.1$ ka. This implies up to 410 m of thinning between 6.5–3.5 ka. Within this period of overall decline there appear two near-instantaneous drops in ice surface elevation of 175 m at $6.5 \pm 0.3$ ka in the Marble Hills, and 210 m at $4.5 \pm 0.4$ ka in the Patriot Hills. No single site contains the entire 410 m thinning signal; the onset is captured best in the Marble Hills while the final stages of thinning are evident in the Patriot Hills.

Our main finding of a mid-Holocene pulse of thinning, complete by 3.5 ka, is insensitive to our interpretation based on the youngest exposure ages. If instead we treat the observed scatter in Holocene exposure ages as a result of both geological processes making some exposure ages too young, and inheritance making some exposure ages too old, we can derive an average thinning rate using all available data[13]. Figure 5 presents a linear regression through the main cluster of Holocene exposure ages to derive average thinning rates for each massif, and for all three sites together on the assumption that thinning occurred concurrently, for example, refs 13,31. The results suggest average thinning rates of $20.9 \pm 2.7$, $6.7 \pm 0.4$ and $5.1 \pm 0.2$ cm a$^{-1}$ (1$\sigma$) or higher in the Marble, Patriot and Independence Hills, respectively. When considering all sites together, the modelling suggests a lower and linear thinning rate of $8.8 \pm 0.2$ cm a$^{-1}$, initiated at about 8.5–9 ka. Using the same statistical approach but including all deglacial exposure ages ( < 15 ka) suggests the onset of thinning occurred no earlier than 9–10.5 ka (Supplementary Fig. 4), in agreement with our youngest-age approach.

## Discussion

The upper limit of little-weathered erratics has previously been used to constrain an ice-sheet model of the LGM ice thickness in the wider Weddell Sea embayment equivalent to 1.4–2 m of global sea level[7,19]. The fit between the model and the geomorphological evidence is best if one assumes that ice streams beneath the present-day Filchner-Ronne Ice Shelf were sliding with little resistance resulting in low surface profiles. The cosmogenic nuclide dating suggests that the ice around the Heritage Range maintained much of its LGM mass until $\sim 6.5$ ka in the mid-Holocene (Fig. 4).

The sudden thinning at 6.5–3.5 ka is argued to coincide with the change from an overall eastward flow to one captured by flow into Horseshoe Glacier. The switch in ice flow direction is best explained by the retreat of the grounding line into Hercules Inlet. At the maximum the eastward flow was towards Institute ice stream and over the Bungenstock Ice Rise, but grounding-line retreat into Hercules Inlet subsequently diverted and lowered the surface of Horseshoe Glacier. This scenario is supported by radar sounding of ice sheet structures that reveals Institute ice stream was affected by flow switching at some time in the Holocene[32,33]. Finally, the mid- to late-Holocene thinning is consistent with both delayed deglaciation and stability or recent thickening in the past few millennia as indicated by global positioning system (GPS)-derived vertical uplift rates and glacial isostatic adjustment modelling[18,27].

The existence of near-maximum ice thicknesses in the inner Weddell Sea embayment in the early Holocene supports the view

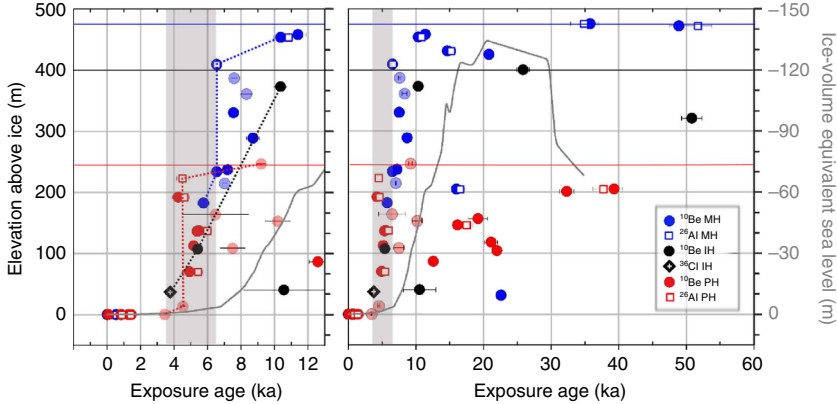

**Figure 4 | The cosmogenic nuclide data.** The apparent exposure ages versus elevation above the present ice surface plotted together with ice-volume equivalent global sea-level changes. The figure shows results for the Marble Hills (MH), Independence Hills (IH) and Patriot Hills (PH) for the Holocene period (left panel) and the past 60 ka (right panel). Error bars (1$\sigma$) reflect analytical uncertainties only. The light-coloured symbols are previously published data[7]. The solid horizontal line is the upper limit of little-weathered erratics for the MH (blue), IH (black) and PH (red). The dotted lines connect the youngest exposure dates at each massif, which are used to infer the mid-Holocene pulse of thinning as indicated by the grey shaded box. The grey line indicates ice-volume equivalent sea-level changes[9]. Our data indicate this sector of the ice sheet contributed to the final stages of sea-level rise.

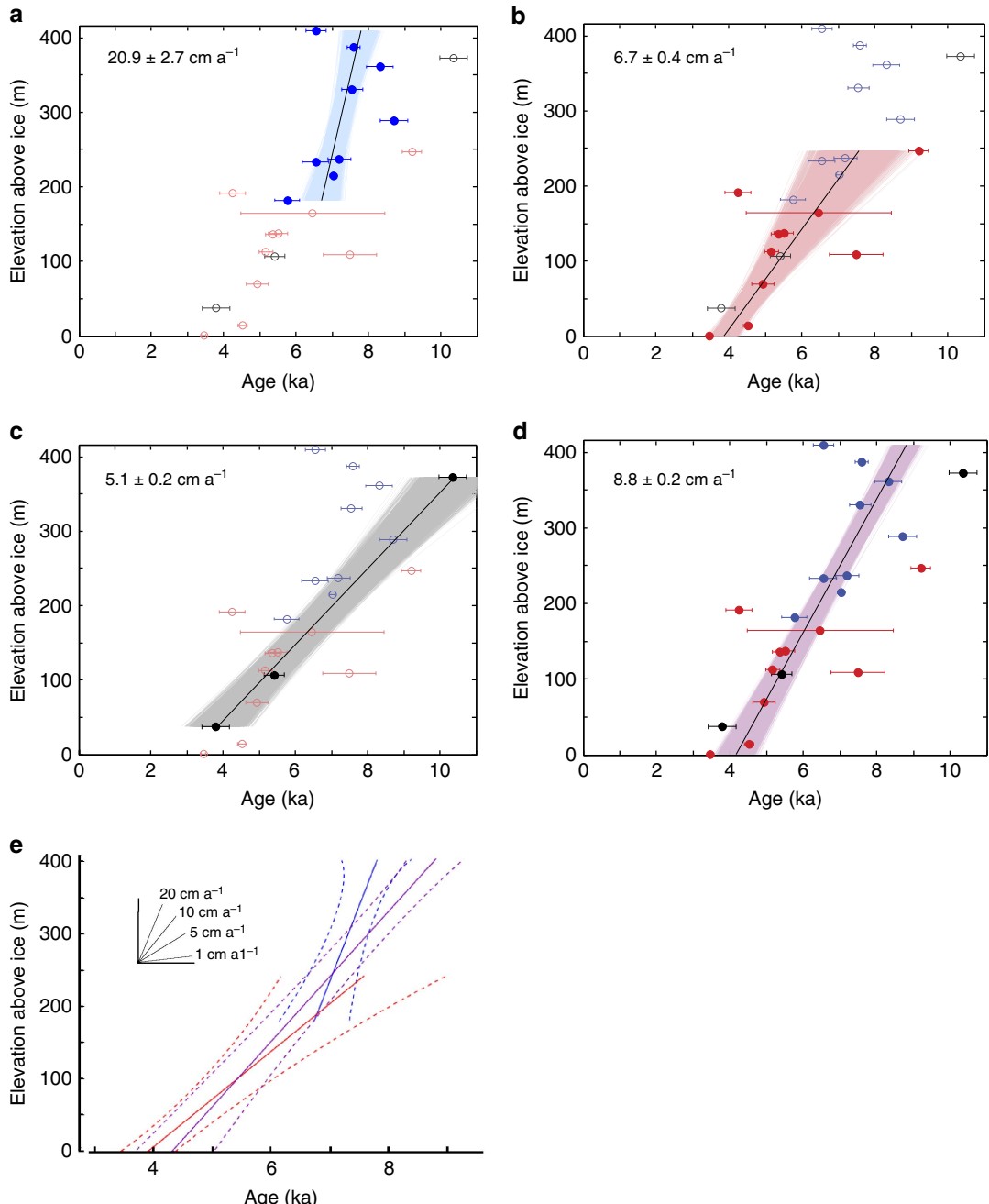

**Figure 5 | Thinning-rate modelling results.** Thinning rates in **a**–**d** are produced using 10,000 iterations of linear regression through the [10]Be data points and their uncertainties, as described in the methods. Uncertainties are 1σ. The data points are coloured to match the different massifs with (**a**) blue from Marble Hills (MH), (**b**) red from Patriot Hills (PH), (**c**) black from Independence Hills, (**d**) and purple for the entire data set. Solid circles within each plot are the data used to model thinning rates. (**e**) A summary of the slopes for MH, PH and the full data set.

that warming following the LGM led to an increase in surface mass balance of the WAIS. This increase in accumulation seems to have compensated for any ice loss through marine downdraw. We argue that the sudden collapse of the ice surface in the Heritage range at 6.5–3.5 ka and accompanying changes in ice flow direction are linked to marine downdraw. The rapidity of decline is a powerful illustration of the dominance of marine downdraw as a mechanism of ice sheet instability in West Antarctica[26].

The evidence of delayed deglaciation from the Atlantic-facing flank of the WAIS following the LGM has implications for global sea level. Rather than contributing to sea-level rise in

the Late Glacial, for example during Meltwater Pulse 1A (14.6–14.3 ka)[6,34], growth of the WAIS interior may have been storing water removed from the oceans at the time. Further, the loss of ice mass persisted well into the mid- to late-Holocene (Fig. 4). Applying glacial models[7,19], it is possible that the inner Weddell Sea zone contributed up to 1.4–2 m of sea-level equivalent between 6.5–3.5 ka. This could be one of the sources of water that causes sea level to rise until late in the Holocene[9].

## Methods

**Geomorphology**. Landforms were mapped in the field using differential GPS and satellite imagery. This formed the basis for detailed work on sediment morphology,

lithology, weathering and cosmogenic nuclide analysis. Selected areas, such as complex debris accumulations were mapped with a terrestrial laser scanner as well as high-resolution vertical aerial photographs, taken from an unmanned aerial vehicle[35,36]. A differential GPS system, mounted on a snowmobile, traversed the ice margin of all three massifs to provide a reference surface from which to normalize sample elevations. Topographic control on the ice margin introduces variability in elevation and therefore uncertainties in the normalization process are estimated at ± 15 m. Supplementary Fig. 1 shows an overview image of the Marble, Independence and Patriot Hills in the southern Heritage Range and the sample locations. Supplementary Fig. 2 illustrates the nature of the glaciated upland surface with photos of individual samples.

**Cosmogenic nuclide analyses.** The cosmogenic nuclide data are presented in Supplementary Data 1–3, and Supplementary Figs 1–4. The data relating to the older, weathered deposits in the Patriot and Marble Hills were the subject of a previous publication[28]; however, we include the $^{26}$Al and $^{10}$Be data set here. In addition, we also include in our data set 11 samples previously reported from the same bedrock spurs[7]; the exposure ages of the latter have been recalculated to be consistent with the present study.

The sampling strategy for cosmogenic nuclides was designed to reduce the chance of nuclide inheritance, and exclude the possibility of nuclide loss through postglacial erosion. We targeted sub-glacially derived clasts with striated surfaces and sub-angular to sub-rounded shapes. We sampled the freshest-appearing, quartz-bearing, brick-sized clasts resting on flat bedrock to minimize problems of post-depositional movement and self-shielding. It is crucial to be convinced that the exposure ages reflect the time since deposition of a freshly exposed clast rather than a signal inherited from the past. To test whether clasts emerging on the glacier surface in blue-ice areas have no inherited cosmogenic nuclides, we analysed seven clasts on the present ice margin. All had relatively low amounts of cosmogenic $^{10}$Be and $^{26}$Al, equivalent to 0–0.8 ka and in two cases 1.5 ka of pre-exposure. In view of the similar lithologies, and thus origin, of quartz-rich erratics at higher elevations, it is reasonable to argue that they too were deposited with pre-exposure of a similar magnitude. On the Independence Hills medial moraine ridge, we collected four large, broad-based limestone and sandstone boulder samples to determine the age of this feature.

The presence of striations on most sampled rock surfaces validates our assumption of negligible rock surface erosion. However, it is worth exploring the sensitivity of our results to this assumption. The effect of incorporating even an exceptionally high Antarctic erosion rate of 2.5 mm ka$^{-1}$ would be to increase LGM ages by ~4% and Holocene ages by 2%. The high end of most Antarctic sandstone erosion rates is ~1 mm ka$^{-1}$; applying this rate would increase LGM ages by 2% and Holocene ages by 1%. Therefore, we argue the results are insensitive to our assumption of zero erosion.

In this study, we favour the youngest exposure age at each altitude to best represent the elevation of the ice surface at the time (Fig. 4; Supplementary Fig. 3). This is a common approach in Antarctica where problems of pre-exposure/ inheritance dominate processes that might cause ages to be too young. For example, we are less concerned with shielding of a sample from cosmic radiation by snow, soil, loess or vegetation in a polar environment. Other forms of shielding such as exhumation through till, or sample self-shielding by rolling of clasts, can be avoided by sampling clasts situated on flat bedrock, as in the present study. One feasible process that could cause erroneously young exposure ages would be in areas of stranded, ice-cored tills. Here a clast could melt out to become exposed after the ice sheet thinned. We avoid this by sampling from bedrock ridges rather than embayments where ice-cored tills are more common. Our seven ice-margin samples act as geological blanks. These suggest that inheritance is low, but could account for up to 1.5 ka of pre-exposure even before being incorporated into the blue-ice moraine. Blue-ice moraines are ice-marginal, supraglacial features and it is reasonable to assume there may be a period of exposure within the blue-ice moraine before the clast is eventually deposited on the mountainside as the ice sheet thins (Supplementary Fig. 2). This is likely to explain much but not the entire observed scatter in the age versus elevation plots, and it reinforces our decision to favour the youngest samples.

Figure 5 explores the effect of being wrong in our assessment of the scatter of exposure ages in terms of pre-exposure and delayed deposition. Rather than preferentially selecting the youngest ages, we treat the observed scatter as a result of both geological processes making some exposure ages too young, and inheritance making some exposure ages too old; in doing so we can derive an average thinning rate using all the available data. In this way, we produce an average thinning rate for each individual massif, and for all three sites together under the assumption that thinning occurred concurrently, for example, refs 13,31. We follow the approach of Johnson et al.[13] in modelling 10,000 linear regressions through all $^{10}$Be exposure ages < 10 ka on each massif. For each iteration, the age for each point is randomly chosen from within the uncertainty bounds on that sample and then a linear regression is fit to those points, removing any results with negative slopes (negligible: <0.001%). Uncertainties are calculated statistically so that 68% of the resulting slope values fall within the given range. The results suggest average thinning rates of 20.9 ± 2.7 cm a$^{-1}$ at Marble Hills, 6.7 ± 0.4 cm a$^{-1}$ at Patriot Hills, and 5.1 ± 0.2 cm a$^{-1}$ at Independence Hills (1$\sigma$), but these rates would be higher following the youngest exposure age approach. When considering all sites together, the modelling suggests a lower and linear thinning rate of

8.8 ± 0.2 cm a$^{-1}$, but initiated a little earlier at about 8.5–9 ka. The exercise demonstrates that our conclusion of a mid-Holocene pulse of thinning, which was complete by 3.5 ka, is not sensitive to our interpretation based on the youngest exposure. In Supplementary Fig. 4, we increase the range to include all $^{10}$Be exposure ages < 15 ka, and then with three clear outliers (3$\sigma$) removed to obtain a linear regression through all deglacial exposure ages. The model suggests the onset of initial deglaciation was at 9–10.5 ka, with a lower average thinning rate of 8.1 ± 0.2 cm a$^{-1}$. The onset of thinning implied is similar to that inferred from the youngest exposure age at Supplementary Fig. 4c, 10 ka.

**Laboratory and analytical techniques.** Whole rock samples were crushed and sieved to obtain the 250–710 μm fraction. Be and Al were selectively extracted from the quartz component of the whole-rock sample at the University of Edinburgh's Cosmogenic Nuclide Laboratory following established methods[37,38]. $^{10}$Be/$^9$Be and $^{26}$Al/$^{27}$Al ratios were measured in 20–30 g of quartz at the Scottish Universities Environmental Research Centre (SUERC) Accelerator Mass Spectrometry (AMS) Laboratory in East Kilbride, UK. Measurements are normalized to the NIST SRM-4325 Be standard material with an assumed[39] $^{10}$Be/$^9$Be of $2.79 \times 10^{-11}$, and the Purdue Z92-0222 Al standard material with a nominal $^{26}$Al/$^{27}$Al of $4.11 \times 10^{-11}$, which agrees with the Al standard material of Nishiizumi[40]. SUERC $^{10}$Be-AMS is insensitive to $^{10}$B interference[41] and the interferences to $^{26}$Al detection are well characterized[42]. Process blanks ($n = 6$) were spiked with 250 μg $^9$Be carrier (Scharlau Be carrier, 1,000 mg l$^{-1}$, density 1.02 g ml$^{-1}$) and 1.5 mg $^{27}$Al carrier (Fischer Al carrier, 1,000 p.p.m.). Samples were spiked with 250 μg $^9$Be carrier and up to 1.5 mg $^{27}$Al carrier (the latter value varied depending on the native Al-content of the sample). Blanks range from $3.3–9.3 \times 10^{-15}$ [$^{10}$Be/$^9$Be] (<1% of total $^{10}$Be atoms in all but the ice-margin samples); and $1.6–7.5 \times 10^{-15}$ [$^{26}$Al/$^{27}$Al] (<1% of total $^{26}$Al atoms in all but the ice margin samples). Concentrations in Supplementary Data 1 are corrected for process blanks; uncertainties include propagated AMS sample/lab-blank uncertainty and a 2% carrier mass uncertainty and a 3% stable $^{27}$Al measurement (ICP-OES) uncertainty.

Cl was extracted from three carbonate samples at the University of Edinburgh following a similar procedure to those outlined in Marrero et al.[43]. A $^{35}$Cl carrier (ORNL batch 150301, ~1.6 mg $^{35}$Cl) was added to the samples before dissolution. The sample was dissolved using trace analysis quality nitric acid and Cl was precipitated as AgCl. The AgCl was purified, pressed and then measured at the SUERC AMS Laboratory. The sample was blank-corrected individually for $^{36}$Cl and total Cl concentrations (~2% each) in the process blanks for that batch of samples. Full sample chemistry and measurement information can be found in Supplementary Data 2.

**Exposure ages.** For exposure age calculations we used default settings in Version 2.0 of the CRONUScalc programme[44]. This is the product of the CRONUS-Earth collaboration that allows for all commonly used nuclides to be calculated using the same underlying framework, resulting in internally consistent cross-nuclide calculations for exposure ages, erosion rates and calibrations. The exposure ages were calculated using a $^{10}$Be half-life of 1.387 Ma (refs 45,46), an $^{26}$Al half-life of 0.705 Ma (ref. 47), and a $^{36}$Cl half-life of 0.301 Ma. The CRONUS-Earth production rates[44,48,49] with the nuclide-dependent scaling of Lifton–Sato–Dunai[50] were used to calculate the ages presented in the paper. Sea level and high latitude production rates are 3.92 ± 0.31 atoms per g per a for $^{10}$Be, 28.5 ± 3.1 atoms per g per a for $^{26}$Al, 56.0 ± 4.1 atoms per (g Ca) per a$^{-1}$ for $^{36}$Cl–Ca and 759 ± 180 neutrons per (g air) per a$^{-1}$ for $^{36}$Cl–P$_f$(0). The sample had very low Cl (15.6 p.p.m.), so the choice of P$_f$(0) does not change the age. The use of Lal/ Stone[51,52] scaling does not change the conclusions of the paper. If Be ages are calculated using the Lal/Stone scaling model, most ages (>1 ka and <1 Ma) range from 1.8 to 3.8% older, with an average of 2.8%. Older samples (>1 Ma) average 4.6% older. For Al samples, most ages range from 5.3 to 8.7% older, with an average of 6.9%; older samples are 11.5% older. The Cl age is 11% older using Lal/Stone scaling. Rock density is assumed 2.7 g cm$^{-3}$ for quartz-bearing erratics and 2.5 g cm$^{-3}$ for limestone; the attenuation length used is 153 ± 10 g cm$^{-2}$. No corrections are made for rock surface erosion or snow cover and thus exposure ages are minimal. Finally, we make no attempt to account for production-rate variations caused by elevation changes associated with glacial isostatic adjustment of the massif through time. Any glacial isostatic uplift would cause the exposure ages to be too young. However, we note that glacial isostatic adjustment in this area is low[27], and the effect would be minimal on the youngest ages.

**Data availability.** The authors declare that the data supporting the findings of this study are available within the article and its Supplementary Information and data files. Any further data or information is available on request from the corresponding author (A.S.H.).

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

## Acknowledgements

The research was funded by the UK Natural Environment Research Council lead grant numbers NE/I025840/1 and NE/I024194/1, and all the data are provided in the Supplementary Materials. We thank the British Antarctic Survey for logistical and field support. We thank three anonymous reviewers for their constructive comments.

## Author contributions

A.S.H., J.W. and D.E.S. conceived the project and carried out the fieldwork and analysis with S.A.D., S.M.M., K.W. and M.J.W. The cosmogenic nuclide analysis was by A.S.H., S.P.H.T.F., R.P.S. and S.M.M. All contributed to the writing of the paper.

## Additional information

**Competing financial interests:** The authors declare no competing financial interests.

