## [Peer review file · Nature Communications]

Reviewers' Comments:

Reviewer #1 (Remarks to the Author)

The potential instability of the West Antarctic Ice Sheet (WAIS) in response to global warming is of wide concern due to the large increase in global sea level that would result from collapse of the ice sheet. Knowledge of WAIS behavior during the warming associated with the last deglaciation provides empirical constraints on potential rates of ice loss and targets for ice sheet models. This manuscript presents new data constraining deglaciation of the Weddell Sea Embayment of West Antarctica. The key result is that the ice surface in this sector of the Weddell Sea Embayment thinned ~ 400 m between (8 -3.5 ka), contributing > 1 m to global sea level during the Middle Holocene. Such data are sparse in this sector of the WAIS and thus this manuscript represents a significant contribution to our knowledge of WAIS deglaciation. As such, the manuscript is appropriate topic for Nature Communications.

The authors use well-established techniques, combining detailed mapping of glacial deposits with surface exposure dating of glacial clasts to constrain WAIS deglaciation. They are aware of well-known challenges to employing surface exposure dating in Antarctica, and minimize potential prior exposure and exhumation by sampling only clasts with evidence of glacial processing, setting on bedrock. The results are interpreted using the youngest ages on simple age-elevation plots. The resulting thinning rates are supported more detailed statistical methods in the Supplement that evaluate the choice of using the youngest ages to infer rates of down draw (Supp figure 5). Regression lines are fit to the data assuming the scatter is normally distributed and modeling 10,000 iterations to generate a distribution of thinning rates. I would like to see this figure incorporated in the main text. Uncertainties on the thinning rates should be included, making the results more robust than the simple empirical methods.

Overall the text is clear and well written, but the authors have a tendency to mix of fact and inference. A typical example occurs in Line 22: the pulse of thinning "triggered by grounding line retreat into Hercules inlet", but the authors do not have any direct constraints on the grounding line position. This phrase is better placed with a suitable qualifier in the last line of the paragraph (line 27). The stepped decline likely reflects marine down draw triggered by grounding line retreat into Hercules inlet.

The figures are informative and generally support the conclusions and interpretations made in the text. It is in fact the comprehensiveness of the figures that leads me to question the conclusion that the ice flow reversed during deglaciation (see below). I also found Supp Fig 5 to be very useful and convincing and would like to see it included in the main body of the manuscript. This could be done if Fig. 2 is removed, or moved to the supplement. Figure 1 could be improved. Fig 1a could be turned into a small inset location map in Fig 1b which would then become the new Fig 1a. The velocity field now shown in Fig 1a would not be visible in the small location map but the area of interest is too small even now. However it could be shown on what is now Fig 1b which would be more useful. I found Supp Fig. 1 very useful while reading the main text; it could be the new Fig 1 b

The references are appropriate and complete. Note Supplementary reference #8, is an out of date reference and the same as #5.

One aspect of the paper that I believe needs to be addressed is the inferred switching of the local ice flow direction from eastward during the LGM to northwestward at the present. This conclusion is based largely on observations of the medial moraine at Independence Hills which appears to be drawn out to the east (lines 79, 239; Fig 2). This interpretation is consistent with the longitudinal

gradient of the medial moraine, the height of the drift at the three massifs and the older evidence of eastward ice flow directions from above the drift limit. It is also consistent with past eastward flow inferred from glaciological studies around Institute Ice Stream and Bungenstock Ice Rise. However, there is no good evidence that the medial moraine was "drawn out" from the large area of blue ice moraine at Independence Hills and the supporting evidence is also consistent with no ice flow reversal.

Instead of being drawn out, the medial moraine appears to be the source of the large concentration of blue ice moraine debris at Independence Hills. As the authors and others have demonstrated elsewhere, blue ice moraines form as englacial debris is exposed as the ice surface ablates by sublimation. Glacial debris is concentrated in blue ice areas by compressive flow and the continued flux of englacial debris from below. As a result, supraglacial debris concentrations increase in the direction of ice flow. The debris concentration in the medial moraine increases toward the southwest (see Fig 2 and supp Fig 1 &2), consistent with the present flow regime. In addition, the medial moraine is contiguous with the outermost (youngest) part of the blue ice moraine. If flow reversal occurred this would imply that no new debris has been exposed since the flow reversal which is presumed to have occurred thousands of year ago, during the drawdown of the ice surface. Other than the superficial appearance, there is no evidence that the moraine formed by drawing out the blue ice moraine. Thus, the medial moraine does not appear to be a relict feature relating to a different ice flow direction.

As the author's previous work shows and the present work assumes, these blue ice areas are long-lived and persisted through the LGM. In order to flow away from the blue ice moraine the area would have to be a local accumulation area, contrary to the other evidence presented. The folding observed within the Independence Hills blue ice moraines (Supp Fig 2) indicates some changes in flow during deglaciation, but does not require reversal of flow. The folding could result in complex exposure age distributions within the blue ice moraine, which may explain some of the scatter in the exposure ages obtained from stranded moraine deposits above (unfortunately no exposure age data is presented from this feature).

The other evidence cited for eastward flow during the LGM is not conclusive. The elevation profile (inset fig 2) of the medial moraine is essentially flat and suggests flow directions perpendicular to the moraine. In any case, the present profile does not constrain the past ice topography when the ice sheet was 100's of meters thicker. The wavy present ice profile likely reflects the very local balance between upward flow and sublimation. For example, the rapid drop off toward A' in the inset of Figure 2 does not represent the regional flow direction.

The small difference (75 m) in relative height of unweathered drift at the Marble and Independence Hills is also consistent with a thicker ice under a flow regime similar to present draining into Horseshoe Glacier as ice elevations are higher there now. Evidence of eastward ice flow of even thicker, older, glaciations is not a good predictor of younger, thinner ice flow given the orientation of the massifs athwart the regional ice flow. The geomorphology of the blue ice moraines is consistent with thinning of relatively flat blue ice areas and recent down draw of ice into Horseshoe Glacier.

Significantly, the inference of a reversal in ice flow direction during deglaciation is not integral to the major conclusions of the paper concerning the timing of Holocene down draw of the WAIS here. Unless additional data supporting ice flow reversal can be shown, for example radar data showing the medial moraine continues to the east rather than swinging to the south, or exposures ages from the supraglacial debris, I recommend that this discussion be dropped from the manuscript.

Other suggestions to improve the clarity of the text.

Line 37-40: This sentence is vague- it could end at uncertain. Otherwise different interpretations need to be spelled out. In that case it would be better as intro to paragraph starting on line 55.

Line 42: higher resolution dating of what? Ice cores or glacial drift?

Line 47: max ice persisted until 12-7 ka. It is not constrained when that elevation was first reached.

Line 50: better- a wave of thinning that propagates to interior. It is important to make clear this is time transgressive and not an immediate response.

Line 63: earlier blue-ice moraines

Line 68: drop "and" replace with ; or end sentence.

Line 72-74: Not convincing, see discussion above.

Line 96-98: this sentence is confusing; better to break into two short sentences.

Line 108: I don't think re-deposition of clasts with prior exposure be ruled out for samples with older exposure ages.

Line 112-120: not convinced of the conclusion of flow switching as discussed above.

I recommend accepting this manuscript subject to revisions outlined above.

Reviewer #2 (Remarks to the Author)

Accurate chronologies of past glaciation in Antarctica are critical for understanding ice-sheet behavior, both past and future, and for ground-truthing ice-sheet models. However, given the size of the continent, reliable chronologies for the Antarctic ice sheet are remarkably sparse. Thus, data such as those presented in this paper, are of great value and aid in our understanding of ice-sheet evolution.

Overall, I have a very favorable impression of this paper. The data are significant and from a region with few prior studies. They bear on a large and important region of the West Antarctic Ice Sheet. They cover a key time period, the deglaciation, and add to a growing body of evidence that Antarctic deglaciation lagged significantly that which occurred elsewhere in the world.

Specific Comments on the text:

L.21-22 The authors suggest ice remained at the LGM position from 49-10 ka. I have doubts about this claim from the data (see below), but if it were true, then the statement earlier in the sentence about being 'nourished by increased snowfall until the early Holocene' would be partially in error, as snowfall only increased during the termination of the ice age (post 20 ka) and was quite low during the LGM.

L .73 It seems reasonable to infer that LGM ice flow was to the east, given the blue ice moraine orientation in the Independence Hills. But, I don't see that this eastward flow is demonstrated by the longitudinal gradient of the moraine, as stated in lines 73 and 74. In fact, in the figure, there doesn't really appear to be a gradient, but rather it looks quite flat, except where the ice has been pirated.

L. 92-99 I agree that the data show rapid ice thinning in the mid-Holocene. However, no one site shows the total drop nor the complete time period 3.5-6.5 ka. Ice level appears to drop much earlier at the Marble Hills than in the Patriot Hills. Ice in the Patriot Hills apparently remains very

close to its maximum well after the rapid drop in the Marble Hills. Why should there be such a significant difference among sites that are so close together? A few sentences to address this issue would be useful.

L. 107-109. Given the problem of inheritance, I have serious doubts about the claim that ice attained its maximum thickness at 50 ka before the global LGM. This is based on a single date. Nowhere in Antarctica would I trust a single cosmogenic date. Yes, the modern rocks in the blue ice zone have near-zero ages. But, that does not mean that rocks in the past did (especially with different flow directions and ice thicknesses) nor does it mean that the glacier did not move around, bury, and/or partially erode rocks already sitting on the landscape. This is a weak conclusion in the paper and detracts from the much stronger story of Holocene deglaciation.

L. 112. There seems to be a mix of fact and inference in this paragraph. Is there independent evidence for the timing of the flow switch? Or is it inferred that this is the most likely time for the flow switch, given that this is when the thinning occurred? There is a subtle difference and a few changes to the language here would help to clarify that this is a likely scenario but not a proven fact.

L. 124-127. These sentences are a bit confusing and seem overly speculative.

L. 138-139. Last sentence is awkward

Supplementary Figure 3 - A - in the photo it is unclear that this is a moraine as opposed to drift patches that may or may not mark the maximum position. The sediment the person is standing on is unconvincing as viewed in the photo. Does the moraine continue in the background of the photo? It almost looks as if there might be a ridge there, but it is hard to tell. Perhaps a little more text would explain the photo better and/or arrows could point to the moraine.

L 313 'nuclide loss through postglacial erosion'. 'Postglacial' is necessary here.

L. 322 See comment above about using blue ice rocks as 'geological blanks'.

L. 340-341 I agree some of the scatter in the data may be caused by exposure first in blue ice zones, but the scatter in the data is too large to be reasonably explained entirely by this scenario and suggests some degree of inheritance.

L. 345 What is the influence of assuming a normal distribution on your results?

Reviewer #3 (Remarks to the Author)

A. Summary of the key results

Hein et al. present new cosmogenic surface exposure ages from the Weddell Sea sector of the Antarctica Ice Sheet (AIS) in order to provide geo-chronologic constraints on the ice sheet history of the past 50,000 years. The paper attempts to address the ice sheet history of this sector of the AIS to provide further constraints on the global sea level history during the last glacial termination. Ages for surface exposure dating were collected from vertical transects in the Heritage Range that range from 700-1400m elevation.

B. Originality and interest: if not novel, please give references

This work builds on prior surface exposure dates from the Heritage Range (Bentley et al., 2010), but now with a substantial increase in the data. From my perspective, this is an interesting dataset and the conclusions by the authors will be of general interest to the larger scientific community who work on glacial histories and sea level changes over the last deglaciation. However, while I think the additional data improves our confidence in the deglacial history of the ice sheet in the Weddell Sea region, I do not see a large improvement or novel result compared to the original

work by Bentley et al. (2010). Specifically, how does this new dataset from Hein et al. significantly improve from what was already gained from Bentley et al. (2010): specifically, I'm referring to Bentley et al.'s Figure 2 and Hein et al.'s Figure 3, which I've attached for general reference. Unless the authors address this issue head on, I find the new data interesting but not novel or a significant improvement from Bentley et al.'s original work.

C. Data & methodology: validity of approach, quality of data, quality of presentation

The authors have applied standard techniques to address their proposed hypothesis - surface exposure dating with the cosmogenically derived isotopes of ^{10}Be , ^{26}Al , and ^{36}Cl . This dating technique is now standard fair in glacial terrains (and sometimes the only available technique) and the authors have addressed and acknowledge the limitations of the technique, which for Antarctica is primarily inheritance from prior surface exposure.

The data appear to be in good shape and the methodologies the authors apply are sound. I suspect some other reviewers may grumble about what exact scaling technique ends up being used and whether choosing the youngest age for surface exposure dating is appropriate. From my perspective, it will not effect their main conclusions and mostly only refine the age ranges slightly. The authors have address this issue as best can be done.

However, I do think the authors need to spend some more time estimating how erosion could effect their ages, given they choose to use the youngest age to best represent their exposure histories. In some situations and locations in Antarctica, erosions rates have been estimated as high as 2-4m/Myr (Summerfield et al., 1999; Putkonen et al., 2008), which could have an appreciable effect on the exposure ages (~10% greater in age). Couldn't the authors use their paired ^{10}Be and ^{26}Al ages to constrain this? Regardless, given many of their samples are sandstone, it seems justifiable for the authors to provide the reader with more information about why they chose an erosion rate of 0cm/yr and how that could influence their conclusions.

D. Appropriate use of statistics and treatment of uncertainties

Yes.

E. Conclusions: robustness, validity, reliability

The conclusions by the authors seems justified based on the data, but I do not find the conclusions a significant improvement from prior work (see comments above).

F. Suggested improvements: experiments, data for possible revision

The authors need to make a better case as to why the new data presented in this paper provides an improvement in the interpretation of the work by Bentley et al. (2010). At the moment, they seem to just confirm what has already been published in the Ellsworth Mountains and nothing more. It is good to replicate data, but simply restating past interpretations does not improve scientific knowledge. What is novel here or how have the new dates improved our understanding of the glacial history? This needs to be tackled head on to improve the understanding of the regional glacial history.

G. References: appropriate credit to previous work?

Yes.

H. Clarity and context: lucidity of abstract/summary, appropriateness of abstract, introduction and conclusions

The clarity and context are appropriate.

Specific Comments:

Abstract:

- What evidence is there for grounding line retreat? Is this simply being speculated on by the dates themselves or are the authors referring to some prior work (perhaps the Bentley et al. paper?). Some clarification in the text would be useful.

- Where do the sea level equivalent numbers (i.e. 1.4-2m) come from? Again, this appears to be from the original work in the area by Bentley et al., but I don't see where the authors provide the source?
- I do not agree with the specific range of the ice elevation fall (6.5-3.5ka) based on the authors own work in the supplemental (e.g. Figure S5). It would appear that the main ice elevation drop occurred somewhere between 9-4ka. If the author provided some justification for choosing the narrower range of 6.5-3.5ka, it would be easier for the reader to determine their reasoning.

Body:

- Line 107-109: I do not think the authors can conclude that the ice reached its maximum thickness based on the available data. If anything, staying true to their 'youngest sample' approach it would appear that ~35ka is a better choice for when the ice attained its maximum thickness, but even then I think this speculative. I suggest to either expand on this in detail or drop the discussion altogether.
- Line 133: Citation needed after Meltwater Pulse 1A
- Line 134: "extracting water" seems like an odd description. Perhaps be more descriptive here and related the accumulation of WAIS to the global sea level lowering.

Supplemental

- Figure S3: These are great photos for getting a sense of the terrain and material dated. It would be even more useful to the reviewer and readers if you showed what some of the samples that provided inherited ages looked like. Are they distinctly different in appearance and/or does the terrain look distinctly different?
- Figure S5, line 371: "e)" should be "d)".
- Supplemental Table 3: The file is very difficult to read on my end. Hopefully, the final version will be easier to read. Although, it would be better if the final version ends up being a .txt or .csv file for archiving of the data and general use later.

The following is a point-by-point response to individual reviewers' comments. There is unavoidable repetition in our response when multiple reviewers have raised similar points. The reviewers' comments are in standard text whilst our responses are emphasized in blue italic text.

Reviewer #1 (Remarks to the Author):

The potential instability of the West Antarctic Ice Sheet (WAIS) in response to global warming is of wide concern due to the large increase in global sea level that would result from collapse of the ice sheet. Knowledge of WAIS behavior during the warming associated with the last deglaciation provides empirical constraints on potential rates of ice loss and targets for ice sheet models. This manuscript presents new data constraining deglaciation of the Weddell Sea Embayment of West Antarctica. The key result is that the ice surface in this sector of the Weddell Sea Embayment thinned ~ 400 m between (8 -3.5 ka), contributing > 1 m to global sea level during the Middle Holocene. Such data are sparse in this sector of the WAIS and thus this manuscript represents a significant contribution to our knowledge of WAIS deglaciation. As such, the manuscript is appropriate topic for Nature Communications.

The authors use well-established techniques, combining detailed mapping of glacial deposits with surface exposure dating of glacial clasts to constrain WAIS deglaciation. They are aware of well-known challenges to employing surface exposure dating in Antarctica, and minimize potential prior exposure and exhumation by sampling only clasts with evidence of glacial processing, setting on bedrock. The results are interpreted using the youngest ages on simple age-elevation plots. The resulting thinning rates are supported more detailed statistical methods in the Supplement that evaluate the choice of using the youngest ages to infer rates of down draw (Supp figure 5). Regression lines are fit to the data assuming the scatter is normally distributed and modeling 10,000 iterations to generate a distribution of thinning rates. I would like to see this figure incorporated in the main text. Uncertainties on the thinning rates should be included, making the results more robust than the simple empirical methods.

We have included this figure in the main text (new Figure 5) and have included uncertainties associated with the modeled thinning rates.

Overall the text is clear and well written, but the authors have a tendency to mix of fact and inference. A typical example occurs in Line 22: the pulse of thinning "triggered by grounding line retreat into Hercules inlet", but the authors do not have any direct constraints on the grounding line position. This phrase is better placed with a suitable qualifier in the last line of the paragraph (line 27). The stepped decline likely reflects marine down draw triggered by grounding line retreat into Hercules inlet.

We have adopted the suggestion of the reviewer and have rewritten the abstract such that the inference is separated from the fact (Line 26-27). Reviewer #2 also raised this point with reference to a sentence now on Line 146; we have adopted these suggestions and separated fact from inference throughout the text.

The figures are informative and generally support the conclusions and interpretations made in the text. It is in fact the comprehensiveness of the figures that leads me to question the conclusion that the ice flow reversed during deglaciation (see below). I also found Supp Fig

5 to be very useful and convincing and would like to see it included in the main body of the manuscript. This could be done if Fig. 2 is removed, or moved to the supplement. Figure 1 could be improved. Fig 1a could be turned into a small inset location map in Fig 1b which would then become the new Fig 1a. The velocity field now shown in Fig 1a would not be visible in the small location map but the area of interest is too small even now. However it could be shown on what is now Fig 1b which would be more useful. I found Supp Fig. 1 very useful while reading the main text; it could be the new Fig 1 b

We appreciate the comments on figures and the possible re-organisation of these within the text. The format for Nature Communications allows us to include additional figures. Therefore, we have adopted the reviewers suggestion of including Supplementary Figures 1 and 5 in the main paper. These figures are now Figures 2 and 5 in the main paper.

The references are appropriate and complete. Note Supplementary reference #8, is an out of date reference and the same as #5.

The duplicate (and out of date) reference has been removed.

One aspect of the paper that I believe needs to be addressed is the inferred switching of the local ice flow direction from eastward during the LGM to northwestward at the present. This conclusion is based largely on observations of the medial moraine at Independence Hills which appears to be drawn out to the east (lines 79, 239; Fig 2). This interpretation is consistent with the longitudinal gradient of the medial moraine, the height of the drift at the three massifs and the older evidence of eastward ice flow directions from above the drift limit. It is also consistent with past eastward flow inferred from glaciological studies around Institute Ice Stream and Bungenstock Ice Rise. However, there is no good evidence that the medial moraine was "drawn out" from the large area of blue ice moraine at Independence Hills and the supporting evidence is also consistent with no ice flow reversal.

Instead of being drawn out, the medial moraine appears to be the source of the large concentration of blue ice moraine debris at Independence Hills. As the authors and others have demonstrated elsewhere, blue ice moraines form as englacial debris is exposed as the ice surface ablates by sublimation. Glacial debris is concentrated in blue ice areas by compressive flow and the continued flux of englacial debris from below. As a result, supraglacial debris concentrations increase in the direction of ice flow. The debris concentration in the medial moraine increases toward the southwest (see Fig 2 and supp Fig 1 & 2), consistent with the present flow regime. In addition, the medial moraine is contiguous with the outermost (youngest) part of the blue ice moraine. If flow reversal occurred this would imply that no new debris has been exposed since the flow reversal which is presumed to have occurred thousands of year ago, during the drawdown of the ice surface. Other than the superficial appearance, there is no evidence that the moraine formed by drawing out the blue ice moraine. Thus, the medial moraine does not appear to be a relict feature relating to a different ice flow direction.

As the author's previous work shows and the present work assumes, these blue ice areas are long-lived and persisted through the LGM. In order to flow away from the blue ice moraine the area would have to been a local accumulation area, contrary to the other evidence presented. The folding observed within the Independence Hills blue ice moraines (Supp Fig 2) indicates some changes in flow during deglaciation, but does not require

reversal of flow. The folding could result in complex exposure age distributions within the blue ice moraine, which may explain some of the scatter in the exposure ages obtained from stranded moraine deposits above (unfortunately no exposure age data is presented from this feature).

The other evidence cited for eastward flow during the LGM is not conclusive. The elevation profile (inset fig 2) of the medial moraine is essentially flat and suggests flow directions perpendicular to the moraine. In any case, the present profile does not constrain the past ice topography when the ice sheet was 100's of meters thicker. The wavy present ice profile likely reflects the very local balance between upward flow and sublimation. For example, the rapid drop off toward A' in the inset of Figure 2 does not represent the regional flow direction.

The small difference (75 m) in relative height of unweathered drift at the Marble and Independence Hills is also consistent with a thicker ice under a flow regime similar to present draining into Horseshoe Glacier as ice elevations are higher there now. Evidence of eastward ice flow of even thicker, older, glaciations is not a good predictor of younger, thinner ice flow given the orientation of the massifs athwart the regional ice flow. The geomorphology of the blue ice moraines is consistent with thinning of relatively flat blue ice areas and recent down draw of ice into Horseshoe Glacier.

Significantly, the inference of a reversal in ice flow direction during deglaciation is not integral to the major conclusions of the paper concerning the timing of Holocene down draw of the WAIS here. Unless additional data supporting ice flow reversal can be shown, for example radar data showing the medial moraine continues to the east rather than swinging to the south, or exposures ages from the supraglacial debris, I recommend that this discussion be dropped from the manuscript.

We appreciate the reviewer's detailed comments on the geomorphological evidence of flow reversal. The reviewer raised some interesting points that need clarification. The Independence Hills blue ice area is unique in the sense that it is surrounded by some of the largest cliffs in the region (>600 m) and thus much of the observed debris on the NW side of the moraine system is rockfall-derived (we are working on a separate publication on rates of rockfall using TLS data) in addition to the folding. Thus, at this site it is not correct to assume the observed concentration of debris here reflects long-term flow direction toward the NW.

We agree that the present-day ice margin profile does not dip in an easterly direction. The inclusion of this profile was meant to show that, at present, there is a clear surface gradient to the N which is in agreement with extensional crevassing, flow structures and drawn out medial moraines between the Independence and Marble Hills that indicate present-day northward flow. We acknowledge that the figure caption mistakenly stated an eastward gradient which is not correct, and indeed the profile is pretty much flat in the central area. What was not clear in the original submission was that the morphology of the medial moraine is consistent with eastward flow. This has now been clarified in the text, which reads "...the addition of ice overspilling from the main WAIS dome displaces the blue-ice moraine and, as is consistent with eastward flow, it forms a medial moraine that is progressively displaced from the mountain foot."

The revised manuscript includes observations and new data to confirm the medial moraine is a relict of former eastward flow. Specifically, the existence of basic igneous rocks at all three

sites that originate in the Scholt Peaks area to the west, and new exposure dates from boulders on the moraine ridge itself, with ages of 17-43 ka. We intended to use these exposure dates from supraglacial moraine material for a publication on the formation process of blue-ice moraines (manuscript in prep), but on the basis of the reviewers comments it seems sensible to include these data in the present manuscript.

We appreciate the reviewer's comments on the issue of flow reversal, and believe the revised manuscript demonstrates overwhelming geomorphological evidence of former eastward flow that changed during the process of thinning. The revised discussion is in the first paragraph of the Results (Line 85).

Other suggestions to improve the clarity of the text.

Line 37-40: This sentence is vague- it could end at uncertain. Otherwise different interpretations need to be spelled out. In that case it would be better as intro to paragraph starting on line 55.

Agreed. The sentence ends at uncertain. We also add a sentence to highlight uncertainty in the area. Lines 38-41.

Line 42: higher resolution dating of what? Ice cores or glacial drift?

This statement has been clarified (Line 43): "latest high-resolution dating of the WAIS Divide ice core..."

Line 47: max ice persisted until 12-7 ka. It is not constrained when that elevation was first reached.

Text changed to read "at its maximum at 12-7 ka". Line 47.

Line 50: better- a wave of thinning that propagates to interior. It is important to make clear this is time transgressive and not an immediate response.

Agreed, we have included this on lines 50-51.

Line 63: earlier blue-ice moraines

Added this (Line 63).

Line 68: drop "and" replace with ; or end sentence.

The sentence was ended (Line 86).

Line 72-74: Not convincing, see discussion above.

This section has been re-written (Lines 85-98)

Line 96-98: this sentence is confusing; better to break into two short sentences.

This sentence has been restructured (Lines 122-124)

Line 108: I don't think re-deposition of clasts with prior exposure be ruled out for samples with older exposure ages.

We have dropped this argument.

Line 112-120: not convinced of the conclusion of flow switching as discussed above.

We have bolstered our reasoning behind this conclusion as detailed above.

I recommend accepting this manuscript subject to revisions outlined above.

Reviewer #2 (Remarks to the Author):

Accurate chronologies of past glaciation in Antarctica are critical for understanding ice-sheet behavior, both past and future, and for ground-truthing ice-sheet models. However, given the size of the continent, reliable chronologies for the Antarctic ice sheet are remarkably sparse. Thus, data such as those presented in this paper, are of great value and aid in our understanding of ice-sheet evolution.

Overall, I have a very favorable impression of this paper. The data are significant and from a region with few prior studies. They bear on a large and important region of the West Antarctic Ice Sheet. They cover a key time period, the deglaciation, and add to a growing body of evidence that Antarctic deglaciation lagged significantly that which occurred elsewhere in the world.

Specific Comments on the text:

L.21-22 The authors suggest ice remained at the LGM position from 49-10 ka. I have doubts about this claim from the data (see below), but if it were true, then the statement earlier in the sentence about being 'nourished by increased snowfall until the early Holocene' would be partially in error, as snowfall only increased during the termination of the ice age (post 20 ka) and was quite low during the LGM.

The terminology has been changed to say the ice "was close to its LGM thickness at 10 ka", thus removing the suggestion that it remained here.

L.73 It seems reasonable to infer that LGM ice flow was to the east, given the blue ice moraine orientation in the Independence Hills. But, I don't see that this eastward flow is demonstrated by the longitudinal gradient of the moraine, as stated in lines 73 and 74. In fact, in the figure, there doesn't really appear to be a gradient, but rather it looks quite flat, except where the ice has been pirated.

Agreed, the original description was not particularly enlightening. We have revised this section to describe the morphology better (Lines 85-98). The sentences state "Eastward flow at the LGM is also consistent with the overall orientation of the blue-ice moraine extending some 10 km to the east of the Independence Hills (Fig. 3C). Here the addition of ice overspilling from the main WAIS dome displaces the blue-ice moraine and, as is consistent with eastward flow, it forms a medial moraine that is progressively displaced from the mountain

foot." The link to the surface gradient was meant to highlight the present northward flow, and this has been clarified in the figure caption and in the following paragraph (Lines 100-109).

L. 92-99 I agree that the data show rapid ice thinning in the mid-Holocene. However, no one site shows the total drop nor the complete time period 3.5-6.5 ka. Ice level appears to drop much earlier at the Marble Hills than in the Patriot Hills. Ice in the Patriot Hills apparently remains very close to its maximum well after the rapid drop in the Marble Hills. Why should there be such a significant difference among sites that are so close together? A few sentences to address this issue would be useful.

Agreed. We now discuss this on lines 126-128, which we suspect is related to local topography and ice supply. During the review process we completed final exposure age results on this project. Included in this final set of dates were two from the altitude profile in the Marble Hills. One was a duplicate from near the upper limit, giving an additional 11 ka age for the timing of initial deglaciation, the second we hoped would fill in the missing gap from low elevation in the Marble Hills. This sample, from bedrock 30 m above the ice margin, gave an inherited age of 22 ka.

L. 107-109. Given the problem of inheritance, I have serious doubts about the claim that ice attained its maximum thickness at 50 ka before the global LGM. This is based on a single date. Nowhere in Antarctica would I trust a single cosmogenic date. Yes, the modern rocks in the blue ice zone have near-zero ages. But, that does not mean that rocks in the past did (especially with different flow directions and ice thicknesses) nor does it mean that the glacier did not move around, bury, and/or partially erode rocks already sitting on the landscape. This is a weak conclusion in the paper and detracts from the much stronger story of Holocene deglaciation.

We have dropped the discussion and now focus on just the Holocene.

L. 112. There seems to be a mix of fact and inference in this paragraph. Is there independent evidence for the timing of the flow switch? Or is it inferred that this is the most likely time for the flow switch, given that this is when the thinning occurred? There is a subtle difference and a few changes to the language here would help to clarify that this is a likely scenario but not a proven fact.

We have altered the sentence as suggested. The sudden thinning "...is argued to coincide with the change..." (Line 145).

S

L. 124-127. These sentences are a bit confusing and seem overly speculative.

This discussion has been dropped from the manuscript.

L. 138-139. Last sentence is awkward

This sentence has been restructured (Lines 166-167).

Supplementary Figure 3 - A - in the photo it is unclear that this is a moraine as opposed to drift patches that may or may not mark the maximum position. The sediment the person is standing on is unconvincing as viewed in the photo. Does the moraine continue in the background of the photo? It almost looks as if there might be a ridge there, but it is hard to

tell. Perhaps a little more text would explain the photo better and/or arrows could point to the moraine.

The image remains but the discussion on the moraine in the main text has been removed.

L 313 'nuclide loss through postglacial erosion'. 'Postglacial' is necessary here.

Added postglacial (Line 191)

L. 322 See comment above about using blue ice rocks as 'geological blanks'.

This is about the best we can do to deal with potential inheritance in rocks. And yes, there still clearly are some rocks with evidence of inheritance.

L. 340-341 I agree some of the scatter in the data may be caused by exposure first in blue ice zones, but the scatter in the data is too large to be reasonably explained entirely by this scenario and suggests some degree of inheritance.

This statement is in reference to the slight (~1.5 ka) scatter in Holocene dates. Of course, there are still some clear outliers outside of this range.

L. 345 What is the influence of assuming a normal distribution on your results?

We favour the youngest exposure age. This exercise is simply to explore whether our conclusions would change significantly if we included all of our data, rather than preferentially selecting the youngest ages. The linear regression forces the normal distribution assumption, right or wrong.

Reviewer #3 (Remarks to the Author):

A. Summary of the key results

Hein et al. present new cosmogenic surface exposure ages from the Weddell Sea sector of the Antarctica Ice Sheet (AIS) in order to provide geo-chronologic constraints on the ice sheet history of the past 50,000 years. The paper attempts to address the ice sheet history of this sector of the AIS to provide further constraints on the global sea level history during the last glacial termination. Ages for surface exposure dating were collected from vertical transects in the Heritage Range that range from 700-1400m elevation.

B. Originality and interest: if not novel, please give references

This work builds on prior surface exposure dates from the Heritage Range (Bentley et al., 2010), but now with a substantial increase in the data. From my perspective, this is an interesting dataset and the conclusions by the authors will be of general interest to the larger scientific community who work on glacial histories and sea level changes over the last deglaciation. However, while I think the additional data improves our confidence in the deglacial history of the ice sheet in the Weddell Sea region, I do not see a large improvement or novel result compared to the original work by Bentley et al. (2010). Specifically, how does this new dataset from Hein et al. significantly improve from what was already gained from Bentley et al. (2010): specifically, I'm referring to Bentley et al.'s Figure 2 and Hein et al.'s Figure 3, which I've attached for general reference. Unless the authors address this issue head on, I find the new data interesting but not novel or a significant

improvement from Bentley et al.'s original work.

Reviewer #3 questions the novelty of this study when compared to the original work in the region by Bentley et al. (2010). The following points aim to clarify the novelty of our contribution.

1) Onset and style of deglaciation. Our new data show the thinning initiated much later than envisaged by Bentley et al (2010), and was not “progressive”. Specifically, we find that the ice margin was within 20 m of the LGM limit at 10 ka, and thus deglaciation initiated more than 5 ka later than indicated in the Bentley et al. (2010) study. Our data also demonstrate that the thinning was not progressive as argued by Bentley et al (2010), but occurred as a “pulse” of accelerated thinning in the mid-Holocene. We have expanded on this point in the manuscript on Lines 72-73 and 119-137.

2) Identification of ice flow reversal and the mechanism causing the pulse of thinning. Another novel contribution of our study is the geomorphological observation of ice flow re-organisation and reversal that accompanies ice sheet thinning. This switch from eastward flow at the LGM to the present northward flow into Hercules Inlet was associated with the pulse of thinning. We link this to a mechanism, marine drawdown, which led to retreat of the grounding line and lowering of the surface of Horseshoe Glacier causing ice to be directed into Hercules Inlet. This is a powerful demonstration of Marine Ice Sheet Instability in the Weddell Sea in the mid-Holocene. We have expanded on this point in the manuscript on Lines 89-98 and 146-154.

3) Robust data set. Here, we provide a robust dataset that offers a high-resolution fix on deglacial thinning in the heart of the Weddell Sea embayment, a significant area representing 25% of Antarctica and an area of high uncertainty. The Bentley et al (2010) study has relatively few dates that constrain the actual deglacial signal since there are many clasts with inheritance. The importance of this is summarised by Reviewer #2 who writes “Nowhere in Antarctica would I trust a single cosmogenic nuclide date”. This comment highlights the need for comprehensive datasets that are unambiguous and that do not rely on just a handful of cosmogenic nuclide dates. Here, we provide such a dataset because we link it to detailed geomorphology and greatly reduce uncertainty on the deglacial history in this part of Antarctica.

4) Identification and explanation for delayed deglaciation. Our discovery that the onset of deglaciation was delayed until the early Holocene is important. This is much later than commonly agreed, for example, in parts of the Antarctic Peninsula Ice Sheet and other areas near the present grounding line. Our paper suggests this results from an increase in surface mass balance of the WAIS following the LGM, as indicated by recent analyses of the WAIS divide core. We suggest a balance between increased snowfall and marine drawdown maintained the elevation until 10 ka.

5) Data to help constrain alternative sea level models. Our data helps to resolve differences between different global sea level curves. For example, some studies suggest sea level had reached present levels by the mid-Holocene at 6 ka (Argus et al., 2014; Mauz et al., 2015), while others show sea level continuing to rise until the late Holocene at 2.5 ka (Lambeck et al., 2014). Our results support this latter view and show a potential source for this water that raised sea level until the late Holocene.

C. Data & methodology: validity of approach, quality of data, quality of presentation
The authors have applied standard techniques to address their proposed hypothesis - surface exposure dating with the cosmogenically derived isotopes of ^{10}Be , ^{26}Al , and ^{36}Cl . This dating technique is now standard fair in glacial terrains (and sometimes the only available technique) and the authors have addressed and acknowledge the limitations of the technique, which for Antarctica is primarily inheritance from prior surface exposure. The data appear to be in good shape and the methodologies the authors apply are sound. I suspect some other reviewers may grumble about what exact scaling technique ends up being used and whether choosing the youngest age for surface exposure dating is appropriate. From my perspective, it will not effect their main conclusions and mostly only refine the age ranges slightly. The authors have address this issue as best can be done. However, I do think the authors need to spend some more time estimating how erosion could effect their ages, given they choose to use the youngest age to best represent their exposure histories. In some situations and locations in Antarctica, erosions rates have been estimated as high as 2-4m/Myr (Summerfield et al., 1999; Putkonen et al., 2008), which could have an appreciable effect on the exposure ages ($\sim 10\%$ greater in age). Couldn't the authors use their paired ^{10}Be and ^{26}Al ages to constrain this? Regardless, given many of their samples are sandstone, it seems justifiable for the authors to provide the reader with more information about why they chose an erosion rate of 0cm/yr and how that could influence their conclusions.

The revised manuscript includes additional justification for why we assume no rock surface erosion. On Line 114 we now specifically state that "rock surface erosion is demonstrably negligible on most of the sampled clasts...". This is then backed up in the Methods section on Lines 193-195 and 207-212. We describe our sampling strategy of targeting striated clasts with subrounded to subangular shapes, and that most clasts demonstrated these features. We now include a discussion on the sensitivity of our results to the effects of erosion. Erosion rates as high as 2-4 mm ka⁻¹ have indeed been reported in Antarctica; however, most studies of Antarctic sandstones report erosion rates of < 1 mm ka⁻¹. Nevertheless, the effect of two different erosion rates is now explored. In the case of a high erosion rate of 2.5 mm ka⁻¹, we find that Holocene exposure ages would increase by 2%. In the case of a lower erosion rate of 1 mm ka⁻¹, the Holocene exposure ages would increase by 1%. Thus, we conclude that erosion would not influence our conclusions.

D. Appropriate use of statistics and treatment of uncertainties
Yes.

E. Conclusions: robustness, validity, reliability
The conclusions by the authors seems justified based on the data, but I do not find the conclusions a significant improvement from prior work (see comments above).

F. Suggested improvements: experiments, data for possible revision
The authors need to make a better case as to why the new data presented in this paper provides an improvement in the interpretation of the work by Bentley et al. (2010). At the moment, they seem to just confirm what has already been published in the Ellsworth Mountains and nothing more. It is good to replicate data, but simply restating past interpretations does not improve scientific knowledge. What is novel here or how have the new dates improved our understanding of the glacial history? This needs to be tackled head on to improve the understanding of the regional glacial history.

See discussion above in Section B.

G. References: appropriate credit to previous work?

Yes.

H. Clarity and context: lucidity of abstract/summary, appropriateness of abstract, introduction and conclusions

The clarity and context are appropriate.

Specific Comments:

Abstract:

- What evidence is there for grounding line retreat? Is this simply being speculated on by the dates themselves or are the authors referring to some prior work (perhaps the Bentley et al. paper?). Some clarification in the text would be useful.

The evidence is discussed on lines 85-109.

- Where do the sea level equivalent numbers (i.e. 1.4-2m) come from? Again, this appears to be from the original work in the area by Bentley et al., but I don't see where the authors provide the source?

The source was cited in the original submission, but now is referred to in the discussion (Lines 142 and 166/7).

- I do not agree with the specific range of the ice elevation fall (6.5-3.5ka) based on the authors own work in the supplemental (e.g. Figure S5). It would appear that the main ice elevation drop occurred somewhere between 9-4ka. If the author provided some justification for choosing the narrower range of 6.5-3.5ka, it would be easier for the reader to determine their reasoning.

There is a full discussion to justify why we favour the youngest exposure age (i.e., the narrower range) in this context on lines 193-229; in particular lines 214-229. We mention why we favour this approach in the main text on lines 116-118 and also point the reader to this more complete discussion in the Methods. We also fully explore and report the alternative interpretation of the data both in the main text (Figure 5, and text on lines 130-137) and in the Methods (lines 231-244).

Body:

- Line 107-109: I do not think the authors can conclude that the ice reached its maximum thickness based on the available data. If anything, staying true to their 'youngest sample' approach it would appear that ~35ka is a better choice for when the ice attained its maximum thickness, but even then I think this speculative. I suggest to either expand on this in detail or drop the discussion altogether.

The discussion has been dropped.

- Line 133: Citation needed after Meltwater Pulse 1A

Citations added.

- Line 134: "extracting water" seems like an odd description. Perhaps be more descriptive here and related the accumulation of WAIS to the global sea level lowering.

This has been reworded to "growth of the WAIS interior may have been storing water removed from the oceans at the time" (Line 165).

Supplemental

- Figure S3: These are great photos for getting a sense of the terrain and material dated. It would be even more useful to the reviewer and readers if you showed what some of the samples that provided inherited ages looked like. Are they distinctly different in appearance and/or does the terrain look distinctly different?

There is no observable difference in the clasts or the depositional surface, certainly not that would be observable in a photograph. We are happy to include additional photos but they would not add much.

- Figure S5, line 371: "e)" should be "d)".

Corrected.

- Supplemental Table 3: The file is very difficult to read on my end. Hopefully, the final version will be easier to read. Although, it would be better if the final version ends up being a .txt or .csv file for archiving of the data and general use later.

We agree. The tables have now been added as Excel worksheet objects that should be accessible in the final version.

Reviewers' Comments:

Reviewer #1 (Remarks to the Author)

This manuscript "Mid-Holocene pulse of thinning in the Weddell Sea sector of the West Antarctic Ice sheet" is a resubmission. The authors have addressed the reviewers concerns with the first version and the manuscript is greatly improved.

The potential instability of the West Antarctic Ice Sheet (WAIS) in response to global warming is of wide concern due to the large increase in global sea level that would result from collapse of the ice sheet. Knowledge of WAIS behavior during the warming associated with the last deglaciation provides empirical constraints on potential rates of ice loss and targets for ice sheet models. This manuscript presents new data constraining deglaciation of the Weddell Sea Embayment of West Antarctica. The key result is that the ice surface in this sector of the Weddell Sea Embayment thinned ~ 400 m between (8 -3.5 ka), contributing > 1 m to global sea level during the Middle Holocene. Such data are sparse in this sector of the WAIS and thus this manuscript represents a significant contribution to our knowledge of WAIS deglaciation. As such, the manuscript is appropriate topic for Nature Communications.

The authors use well-established techniques, combining detailed mapping of glacial deposits with surface exposure dating of glacial clasts to constrain WAIS deglaciation. They are aware of well-known challenges to employing surface exposure dating in Antarctica, and minimize potential prior exposure and exhumation by sampling only clasts with evidence of glacial processing, setting on bedrock. The results are interpreted using two methods. The first utilizes the youngest exposure ages on simple age-elevation plots. This is the method typically used by other authors and is accepted by the community. The weakness of this method is that conclusions are often based on very few samples and high elevation samples with low exposure have strong leverage on the resultant thinning rates and timing of deglaciation. If these samples do not have a simple exposure history as assumed, then the conclusions will be erroneous.

The authors also employ an alternative method (after Johnson et al 2014), that fits a linear regression to the samples < 10 ka to altitude. This cut off age is rather arbitrary but includes all the ages that comprise the sloping cluster of ages that suggest a more or less linear thinning rate (Fig 4). However, this choice insures that the result will not differ significantly from the results based on the youngest ages. It would be better if all ages less than < 16 ka were included as this would test the possibility that deglaciation occurred earlier, as has been inferred in other areas of Antarctica. By inspection of figures 4 and 5, this larger age range would not significantly change the results, except in the Marble Hills and would make the thinning rate there more in line with that at the Patriot Hills and Independence Hills. The higher cut off would make the conclusions much more robust. By its nature, applying a linear regression method eliminates the need for a more rapid pulse of thinning within the deglaciation. Some discussion is in order as to which method is more accurate representation of the glacial history. I would suggest that given the leverage of single samples in the youngest age approach, this statistical method is a more robust method for determining the initiation of deglaciation and thinning rates and should be given priority. Another minor problem is the statement "we assume the observed scatter is normally distributed" (line 230). This statement is very vague (scatter of what about what?) and does not accurately describe the assumptions of the linear regression analysis. Another aspect of this approach is that some samples become outliers that are more than two standard deviations away from the regression line. If true, this implies that in spite of the efforts made in the field, some samples still have complex exposure histories or that thinning was not linear, that is the rate was not constant. This seems plausible and should be discussed.

Overall the text is clear and well written; the authors have significantly improved the revised text.

The figures also have been improved and are informative and support the conclusions and interpretations made in the text. The location of Scholt Peaks should be added to Fig 1B. The references are appropriate and complete.

In the previous version of the manuscript, the evidence as presented for ice flow reversal from eastward during the LGM to northwestward was weak. The authors have improved the discussion and included additional data (erratic provenance, exposure ages) supporting the inferred switching of the local ice flow direction. Although I remain somewhat skeptical based on the morphology of the moraine, the authors present enough evidence to support past eastward flow to justify their inferences. Although the switch is a potential test for ice sheet models, the timing and thinning rates presented are not dependent on whether or not the switching of ice flow occurred. Certainly, the formation of crevasses within the blue ice moraines attests to recent changes in the local flow regime. The formation of blue ice moraines is not completely understood and is focus of current research. The data presented here is a significant contribution to that effort.

In summary, this manuscript is suitable for Nature Communications and I recommend publishing with minor revisions.

Other suggestions to improve the clarity of the text.

Line 23: perhaps substitute thinning rates from linear regression, see discussion above.

Line 33: change "and how long it contributed" to "and its contribution"

Line 37: change "Volumes" to "Ice volumes".

Line 49: change "expansion" to "interior thickening". There is no evidence that the grounding line advanced or that the aerial extent increased.

Line 64: "Little-weathered blue-ice deposits extend 230-475 m above the ice surface" This sentence is repeated more or less below with more detail. As it stands the large range in elevation is confusing. "Little-weathered blue-ice deposits in extend up to 475 m above the ice surface"

Line 75: change to "At the Patriot Hills, the ice elevation today is the same as at 3.5 ka"

Line 85: change "veneer of glacially abraded and often striated" to "veneer of glacial debris including glacially abraded and often striated". Although such glacial processed clasts may be common, I have never seen a deposit where they all are.

Line 92: Add location of Scholt Peaks to Fig 1B and refer to fig here.

Line 131 Unclear what "the observed scatter in Holocene exposure ages is normally distributed" means. See discussion above.

Line 142: with little resistance resulting in low surface gradients. (or low profiles)

Line 143 cosmogenic nuclide dating suggests that the ice around the Heritage range maintained much of its LGM mass until ~6.5 ka

Line 156: Snow accumulation alone is unlikely to compensate marine drawdown which is controlled by conditions at the grounding line. It takes time for the wave of thinning to propagate inland. Apparently it didn't reach the area until ~8 ka. I suppose that high accumulation rates could have buttressed and expanded Skytrain ice rise. In any case, lacking model results this is speculation.

Line 204: The presence of striations on most sampled rock surfaces.

Line 228-248: This section needs to be expanded, perhaps in the supplement (see comments above). If you did get some negative slopes that were discarded why is your distribution not skewed as it was in Johnson et al? In that case, symmetric uncertainties are incorrect. Possibly this only applies to Marble Hills data? Including age up to 16ka might fix this problem.

Reviewer #2 (Remarks to the Author)

I reviewed the original version of this paper. In my opinion, this paper presents key data that bear on the deglacial history of an important sector of the West Antarctic Ice Sheet. The authors have taken the reviewers' comments constructively and made a number of improvements to the paper. All of my comments were addressed, and I have no further suggestions on the paper. With regard to the comments of another reviewer on the overlap with the Bentley et al. paper - I do agree that there is some overlap in that both are from the same general area and cover the same general time period, but the dataset we are reviewing here presents key new and detailed information on the timing and rate of Holocene deglaciation that yields insight that cannot come from the Bentley et al. paper.

Reviewer #3 (Remarks to the Author)

The primary critique of the manuscript related to the novelty of the work. I think the authors have provided a clear explanation as to why these new data help better elucidate the timing of retreat compared to prior work. They carefully considered all of my comments (some of which as I read them now were not well crafted) and the other reviewers. I would suggest publication of the paper. This will certainly be of broad interest to a large number of the Nature Climate Change readership.

The following is a point-by-point response to individual reviewers' comments. The reviewers' comments are in standard text whilst our responses are emphasized in blue italic text.

Reviewer #1 (Remarks to the Author):

This manuscript "Mid-Holocene pulse of thinning in the Weddell Sea sector of the West Antarctic Ice sheet" is a resubmission. The authors have addressed the reviewers concerns with the first version and the manuscript is greatly improved.

The potential instability of the West Antarctic Ice Sheet (WAIS) in response to global warming is of wide concern due to the large increase in global sea level that would result from collapse of the ice sheet. Knowledge of WAIS behavior during the warming associated with the last deglaciation provides empirical constraints on potential rates of ice loss and targets for ice sheet models. This manuscript presents new data constraining deglaciation of the Weddell Sea Embayment of West Antarctica. The key result is that the ice surface in this sector of the Weddell Sea Embayment thinned ~ 400 m between (8 -3.5 ka), contributing > 1 m to global sea level during the Middle Holocene. Such data are sparse in this sector of the WAIS and thus this manuscript represents a significant contribution to our knowledge of WAIS deglaciation. As such, the manuscript is appropriate topic for Nature Communications.

The authors use well-established techniques, combining detailed mapping of glacial deposits with surface exposure dating of glacial clasts to constrain WAIS deglaciation. They are aware of well-known challenges to employing surface exposure dating in Antarctica, and minimize potential prior exposure and exhumation by sampling only clasts with evidence of glacial processing, setting on bedrock. The results are interpreted using two methods. The first utilizes the youngest exposure ages on simple age-elevation plots. This is the method typically used by other authors and is accepted by the community. The weakness of this method is that conclusions are often based on very few samples and high elevation samples with low exposure have strong leverage on the resultant thinning rates and timing of deglaciation. If these samples have do not have a simple exposure history as assumed, then the conclusions will be erroneous.

The authors also employ an alternative method (after Johnson et al 2014), that fits a linear regression to the samples < 10 ka to altitude. This cut off age is rather arbitrary but includes all the ages that comprise the sloping cluster of ages that suggest a more or less linear thinning rate (Fig 4). However, this choice insures that the result will not differ significantly from the results based on the youngest ages. It would be better if all ages less than <16 ka were included as this would test the possibility that deglaciation occurred earlier, as has been inferred in other areas of Antarctica. By inspection of figures 4 and 5, this larger age range would not significantly change the results, except in the Marble Hills and would make the thinning rate there more in line with that at the Patriot Hills and Independence Hills. The higher cut off would make the conclusions much more robust. By its nature, applying a linear regression method eliminates the need for a more rapid pulse of thinning within the deglaciation. Some discussion is in order as to which method is more accurate representation of the glacial history. I would suggest that given the leverage of single samples in the youngest age approach, this statistical method is a more robust method for determining the initiation of deglaciation and thinning rates and should be given priority. Another minor problem is the statement "we assume the observed scatter

is normally distributed" (line 230). This statement is very vague (scatter of what about what?) and does not accurately describe the assumptions of the linear regression analysis. Another aspect of this approach is that some samples become outliers that are more than two standard deviations away from the regression line. If true, this implies that in spite of the efforts made in the field, some samples still have complex exposure histories or that thinning was not linear, that is the rate was not constant. This seems plausible and should be discussed.

The reviewer is suggesting we prioritize the statistical method of determining thinning rates and the initiation of deglaciation, given the leverage of single samples in the 'youngest exposure age' approach. However, we discuss at length our reasoning for adopting the youngest-age approach in the Methods section. We feel this gives us a more accurate representation of the timing and style of deglaciation. However, we acknowledge that taking this approach comes with its own uncertainties, and this is why we have explored the sensitivity of our results using the statistical model. The present manuscript presents our preferred interpretation (i.e., deglaciation initiated at c. 10.5 ka, then rapid thinning 6.5-3.5 ka) and also the statistical interpretation for the timing of the "pulse" of thinning (i.e., initiated at 8.5-9 ka and completed by 4 ka).

The suggestion here calls for accepting all exposure ages less than 16 ka as a means of constraining the onset of thinning and deriving average thinning rates. This approach would include samples with clear inheritance (for example, the 16 ka sample at 200 m above the ice surface at MH). Instead, what we have done is include all samples < 15 ka (following Bentley et al.'s "progressive thinning since 15 ka") and removed the obvious outliers such as the one mentioned above. The result is an average thinning rate of 8.1 +/- 0.2 cm a⁻¹, initiated at 9.2-10.5 ka. As the reviewer suspected, this doesn't change the results much, except the average thinning rate is a little slower. The model suggests the onset of initial deglaciation is similar to our estimate based on the youngest age approach (c. 10.4 ka). We include this as Supplementary Figure 4, with Fig. 4A being the regression without outliers removed, and Fig. 4B the regression with outliers removed.

The model produces a simple, linear thinning history. This is probably unrealistic for an area where thinning is likely controlled by retreat of grounded marine ice, which probably goes through cycles of unstable retreat between stable pinning points.

Overall the text is clear and well written; the authors have significantly improved the revised text. The figures also have been improved and are informative and support the conclusions and interpretations made in the text. The location of Scholt Peaks should be added to Fig 1B. The references are appropriate and complete.

We have included the location of Soholt Peaks in Figure 1B.

In the previous version of the manuscript, the evidence as presented for ice flow reversal from eastward during the LGM to northwestward was weak. The authors have improved the discussion and included additional data (erratic provenance, exposure ages) supporting the inferred switching of the local ice flow direction. Although I remain somewhat skeptical based on the morphology of the moraine, the authors present enough evidence to support past eastward flow to justify their inferences. Although the switch is a potential test for ice sheet models, the timing and thinning rates presented are not dependent on whether or not the switching of ice flow occurred. Certainly, the formation of crevasses within the blue

ice moraines attests to recent changes in the local flow regime. The formation of blue ice moraines is not completely understood and is focus of current research. The data presented here is a significant contribution to that effort.

In summary, this manuscript is suitable for Nature Communications and I recommend publishing with minor revisions.

Other suggestions to improve the clarity of the text.

Line 23: perhaps substitute thinning rates from linear regression, see discussion above.

We have kept our results from the youngest-age approach; our reasoning is well justified. In any case, the key results are not affected (onset of deglaciation in the early Holocene with the bulk of thinning in the mid-Holocene).

Line 33: change "and how long it contributed" to "and its contribution"

The suggested change has been made.

Line 37: change "Volumes" to "Ice volumes".

The suggested change has been made.

Line 49: change "expansion" to "interior thickening". There is no evidence that the grounding line advanced or that the aerial extent increased.

The suggested change has been made.

Line 64: "Little-weathered blue-ice deposits extend 230-475 m above the ice surface" This sentence is repeated more or less below with more detail. As it stands the large range in elevation is confusing. "Little-weathered blue-ice deposits in extend up to 475 m above the ice surface"

The suggested change has been made.

Line 75: change to "At the Patriot Hills, the ice elevation today is the same as at 3.5 ka'

The suggested change has been made.

Line 85: change "veneer of glacially abraded and often striated" to "veneer of glacial debris including glacially abraded and often striated". Although such glacial processed clasts may be common, I have never seen a deposit where they all are.

The suggested change has been made, although at this site the majority of clasts exhibit these characteristics.

Line 92: Add location of Scholt Peaks to Fig 1B and refer to fig here.

The suggested change has been made.

Line 131 Unclear what "the observed scatter in Holocene exposure ages is normally distributed" means. See discussion above.

We have changed the text to be more accurate. It now reads "If instead we treat the observed scatter in Holocene exposure ages as a result of both geological processes making some exposure ages too young, and inheritance making some exposure ages too old, we can derive an average thinning rate using all available data"

Line 142: with little resistance resulting in low surface gradients. (or low profiles)

The suggested change has been made.

Line 143 cosmogenic nuclide dating suggests that the ice around the Heritage range maintained much of its LGM mass until ~6.5 ka

The suggested change has been made.

Line 156: Snow accumulation alone is unlikely to compensate marine drawdown which is controlled by conditions at the grounding line. It takes time for the wave of thinning to propagate inland. Apparently it didn't reach the area until ~8 ka. I suppose that high accumulation rates could have buttressed and expanded Skytrain ice rise. In any case, lacking model results this is speculation.

Speculation is inherent in the terminology used "the increase in accumulation seems to have compensated for ..."

Line 204: The presence of striations on most sampled rock surfaces.

The suggested change has been made.

Line 228-248: This section needs to be expanded, perhaps in the supplement (see comments above). If you did get some negative slopes that were discarded why is your distribution not skewed as it was in Johnson et al? In that case, symmetric uncertainties are incorrect. Possibly this only applies to Marble Hills data? Including age up to 16ka might fix this problem.

The total number of negative slopes was less than 0.001 % of all simulations (this has been added to the text), and only with respect to the <10 ka Marble Hills dataset.

Reviewer #2 (Remarks to the Author):

I reviewed the original version of this paper. In my opinion, this paper presents key data that bear on the deglacial history of an important sector of the West Antarctic Ice Sheet. The authors have taken the reviewers' comments constructively and made a number of improvements to the paper. All of my comments were addressed, and I have no further suggestions on the paper. With regard to the comments of another reviewer on the overlap with the Bentley et al. paper - I do agree that there is some overlap in that both are from the same general area and cover the same general time period, but the dataset we are reviewing here presents key new and detailed information on the timing and rate of Holocene

deglaciation that yields insight that cannot come from the Bentley et al. paper.

Reviewer #3 (Remarks to the Author):

The primary critique of the manuscript related to the novelty of the work. I think the authors have provided a clear explanation as to why these new data help better elucidate the timing of retreat compared to prior work. They carefully considered all of my comments (some of which as I read them now were not well crafted) and the other reviewers. I would suggest publication of the paper. This will certainly be of broad interest to a large number of the Nature Climate Change readership.